# Renal Tumors with Oncocytic and Papillary Features: A Phenotypic and Genotypic Study

**DOI:** 10.3390/diagnostics11020184

**Published:** 2021-01-28

**Authors:** Tania Franceschini, Francesca Giunchi, Veronica Mollica, Annalisa Altimari, Elisa Capizzi, Mattia Banfi, Riccardo Schiavina, Michelangelo Fiorentino, Francesco Massari

**Affiliations:** 1Department of Pathology, IRCCS Azienda Ospedaliero Universitaria di Bologna, Via Albertoni 15, 40138 Bologna, Italy; tania.franceschini@studio.unibo.it (T.F.); frachikka@virgilio.it (F.G.); annalisa.altimari@aosp.bo.it (A.A.); elisa.capizzi@libero.it (E.C.); mattiabanfi12@gmail.com (M.B.); 2Department of Medical Oncology, IRCCS Azienda Ospedaliero Universitaria di Bologna, Via Albertoni 15, 40138 Bologna, Italy; veronica.mollica7@gmail.com (V.M.); fmassari79@gmail.com (F.M.); 3Department of Urology, IRCCS Azienda Ospedaliero Universitaria di Bologna, Via Albertoni 15, 40138 Bologna, Italy; rschiavina@yahoo.it; 4Department of Specialistic Diagnostic and Experimental Medicine, University of Bologna, Via Massarenti 9, 40138 Bologna, Italy

**Keywords:** renal cell tumors, oncocytic tumors, papillary tumors, immunohistochemistry, fluorescence in situ hybridization, next generation sequencing

## Abstract

The occurrence of kidney oncocytic lesions with an admixed papillary component is not unusual in routine pathology practice. These neoplasms with dual morphology are classically recognized as collision tumors with variable malignant potential. Using immunohistochemistry, we investigated fluorescent in situ hybridization and next generation sequencing of the genetic and phenotypic profiles in the two components of 11 kidney tumors with colliding oncocytic and papillary features. The oncocytic component was CD117 positive, CK7 negative, and AMACR negative; the papillary component was CK7 positive, AMACR positive, and CD117 negative in all cases. Fluorescence in situ hybridization (FISH) results were inconsistent. Next generation sequencing (NGS) analysis demonstrated that the mutations identified in the two tumor components were identical and displayed an allelic frequency of approximately 50%, strongly suspicious for genetic polymorphisms. The two oncocytic and papillary tumor counterparts shared the same genetic profile and did not harbor pathogenic mutations. Clinical confirmation of the biological benign features of these tumors is required. The term collision tumor is not suitable for these neoplasms, and we propose the term oncopapillary tumor for this histological entity.

## 1. Introduction

The number of histological new entities among renal cell tumors is steadily increasing over the years [1]. This new landscape of renal cell tumors includes several lesions with eosinophilic or oncocytic histology. Eosinophilic renal neoplasms comprise a spectrum of tumors ranging from the benign oncocytoma to the indolent Hybrid Oncocytic Tumor (HOT) until the frankly malignant Chromophobe Renal Cell Carcinoma (ChRCC) [2,3].

In routine pathology practice, it is not unusual to encounter oncocytic lesions admixed with a papillary component, especially in the central scar of these tumors. The cooccurrence of a tubulo-papillary tumor component besides an oncocytic counterpart in the histological view is classically recognized as a collision tumor [4]. The term collision tumor refers to the presence of coexistent but independent tumors that are histologically distinct [5]. Collision tumors are believed to result from two separate but adjacent neoplasms (biclonal malignant transformation) and are different from composite tumors, which are thought to arise from a multidirectional differentiation of a single neoplasm [6]. Collision tumors imply the coexistence of two discrete histogenetically and genetically distinct cell types arising from a common source.A recent study reported the immunohistochemical (IHC) and the fluorescence in situ hybridization (FISH) characteristics of 17 tumors of the kidney with a papillary component admixed with another histotype [7]. Most neoplasms in this study were papillary renal cell proliferations arising in the background of oncocytoma and chRCC, and the two components showed distinct ICH and FISH features, confirming the collision nature of the tumors [7]. However, the presence of copy number gainsin chromosomes 7/17 has been described also in benign papillary adenomas [8]. The recognition of the malignant histology of at least one component leads to the classification of the collision tumor as malignant with clinical and follow-up consequences.

The genetic background of the papillary and oncocytic components in these tumors and their potential different clonal origins have never been explored so far. Therefore, in this study, we aimed to investigate with a next generation sequencing (NGS) approach the genetic variations in the two components of 11 kidney tumors with putative colliding oncocytic and papillary neoplasms and to compare it with the immunohistochemical and the FISH characteristics.

## 2. Material and Methods

We retrospectively selected 11 patients with diagnosis of renal neoplasia with oncocytic features associated to papillary proliferation diagnosed from 2015 to 2020 at the pathology department of the S.Orsola-Malpighi Hospital (Bologna, Italy). The study was approved by the Ethical Committee of the Area Vasta Centrale Emilia Romagna with the code PRIORI 321/2019/Oss/AOUBo, on 25 June 2019. Six patients (55%) were females, and four were (45%) males; the mean age was 57.6 ± 11.35 years (range 42–80) (Table 1). All tumors had been diagnosed as collision tumors in the original pathology report. In 6 patients, the collision was between hybrid oncocytic tumor and papillary renal cell carcinoma type 1. In 5 patients, the association was between oncocytoma and papillary renal cell carcinoma type 1.

All lesions were histologically re-reviewed by three dedicated uropathologists (T.F., F.G., and M.F.), blinded to the original pathology report, and were classified according to the 2013 International Society of Urological Pathology (ISUP) classification [9].

The surgical specimens were originally formalin-fixed, paraffin-embedded, and routinely processed for histological diagnosis. Three micrometerthick sections were cut from representative paraffin blocks and stained with hematoxylin and eosin, while 4 µm thick sections were prepared for IHC and fluorescent in situ hybridization (FISH) analyses.

Immunohistochemistry was accomplished in all cases using the OptiView DAB IHC Detection Kit on an automated Benchmark Ultra instrument (Ventana Medical Systems, Tucson, AZ, USA). We utilized the following pre-diluted antibodies: cytokeratin 7 (clone SP52), alpha-methyl CoA racemase-AMACR (clone P504S), Ki-67 (clone 30-9), and CD117/CKIT (clone YR145). The following IHC algorithm was applied to diagnose histological components according to the ISUP recommendations [10]: we performed IHC for CD117 for the detection of the oncocytic lesion, either benign or malignant, and CK7 for the differential diagnosis between oncocytoma and chRCC; co-expression of CK7 and racemase was utilized for the diagnosis of the papillary component; the proliferation index Ki67 was utilized with an arbitrary cutoff of 12% to discriminate between oncocytoma and hybrid oncocytic tumors.

FISH for the ploidy (gain or loss) of chromosomes 1,6, 7, and 17 was utilized to identify papillary renal cell carcinoma type I and chRCC as we previously described [2]. We utilized centromeric DNA probes for chromosome 1 (CEP 1, Spectrum Orange), chromosome 6 (CEP 6, Spectrum Green), chromosome 7 (CEP 7, Spectrum Green), and chromosome 17 (CEP 17, Spectrum Orange), all from Abbott Molecular, (Abbott Park, IL, USA).

Slides were evaluated with an epi-fluorescence microscope (Nikon Eclipse 80; Nikon Corporation, Tokyo, Japan) equipped with single band-pass filters. For each sample, 80–100 neoplastic nuclei were analyzed under high-power magnification (1000×). The cutoff values for the definition of chromosomal gains and losses were set at the mean ±3 SD of the control values (nonneoplastic cells). Any tumor with a signal score beyond the cutoff value was considered to have gained or loss that chromosome.

NGS analysis of the two components, oncocytic and papillary of each tumor, was performed after selection and manual microdissection. Tumor areas of interest with at least 70% tumor cell enrichment were circled, and 10 μm thick serial sections of the same paraffin block were cut in sterility for DNA extraction. The DNA was extracted using the GeneRead DNA FFPE Kit (Qiagen, Hilden, Germany) and quantified with the Quantifiler^®^ Human DNA Quantification Kit (Thermo Fisher Scientific, Waltham, MA, USA), and 10 ng of DNA was used for library preparation. Next generation sequencing (NGS) analysis was run on an Ion GeneStudio S5™ System (ThermoFisher Scientific, Waltham, MA, USA) using the Oncomine Comprehensive Assay V3 (ThermoFisher Scientific, Waltham, MA, USA), covering 178 cancer-related genes (87 hotspots, 48 full-length, and 43 copy numbers).

Successful sequencing of a sample required at least 500,000 reads with a quality score ≥ Q20. A minimum coverage of 500× with at least 10% frequency was used as the cutoff for a variant to be considered true. Sequence alignment and base calling were performed using the Torrent Suite software v.4.4.3 (Thermo Fisher Scientific, Waltham, MA, USA) taking Human Genome Build 19 (hg19) as the reference. Variant calling was carried out with the Variant Caller v.4.4.3.3 plug-in, using default “Somatic—Low Stringency” settings. Variants were further filtered using Ion Reporter software v.4.4 (Thermo Fisher Scientific, Waltham, MA, USA).

## 3. Results

The histological review of the lesions revealed that all the cases harbored dual oncocytic and papillary components and that the diagnosis was concordant among the three dedicated genitourinary pathologists.

The IHC profile was in agreement with the histological diagnosis in all cases for each component: the oncocytic component was CD117 positive, CK7 negative, and AMACR negative; the papillary component was CK7 positive, AMACR positive, and CD117 negative. The Ki67 proliferative index was invariably <5% in the oncocytic component and <1% in the papillary counterpart of all lesions (Figure 1).

The FISH analysis demonstrated in 3/10 cases a gain of chromosome 7/17 in the papillary component, as expected in a papillary renal tumor. Concomitant deletion of chromosomes 1 and 6 was observed in one case in both the papillary and the oncocytic components as well as gain of chromosome 1 on both components of another case (Table 2). The other 6 cases turned out diploid in each component at least for chromosomes 1, 6, 7, and 17 (Figure 1).

The NGS analysis was accomplished only in 7 patients since, in three cases, the minimum threshold of tumor cell enrichment (10%) was not achieved in the two tumor components. Sequencing analysis demonstrated that the mutations identified in the two tumor components were identical and displayed an allelic frequency of approximately 50% in all 7 cases. This finding was strongly suspicious for genetic polymorphism rather than pathogenic variants.

The clinical, histological, and molecular features of the cases are described in Table 1 and Table 2.

## 4. Discussion

The occurrence of a papillary component within renal oncocytic tumors is often underestimated and, in case, simply regarded as a collision tumor. This type of renal collision tumors has been widely described in case reports as histological findings [5,11]. A recent report investigated the phenotypic and cytogenetic features of these lesions, concluding that the two histological counterparts are different and should be considered colliding neoplasms [7]. Based on the histological classification, these tumors are composed of a benign part (oncocytic lesion) and a low-grade malignant part (papillary component).

Our results confirm that the histological and the immunohistochemical traits of the oncocytic and the papillary parts of these tumors are morphologically and phenotypically different. However, our study demonstrates that the two tumor counterparts are genetically identical and belong to the same proliferation from a biological point of view. Collision tumors represent a biological enigma since two completely different histological counterparts of a single tumor are clearly separated [12]. The hypothesis of two separate neoplastic clones deriving from different cancer stem cells or the occurrence of an epithelial to mesenchymal transition isthe prevalent explanation for this rare phenomenon that may happen in many organs [11,12,13,14,15]. However, this hypothesis is hardly applicable to benign or low-grade malignant tumors such as the oncocytic and papillary renal neoplasms. Since we demonstrated that the cells of the two tumor counterparts share the same genetic profile, the term collision tumor is not suitable. We may postulate that, in these neoplasms, the cells acquire a different phenotype (expression of CK7, racemase, and CD117) and a diverse morphology due to changes in the tumor microenvironment and to tumor–stroma interactions. In fact, the papillary component almost invariably grows at the interface between the oncocytic cells and the edematous myxoid or hyalinized stroma that is typical of oncocytoma. Since these tumors seem to be a separate histological entity, we propose to call them oncopapillary tumors.

Our FISH results are also inconsistent with typical cancer alterations in most of the tumors of our series. In the papillary component, the presence in some cases of the gain of chromosome 7/17 was actually described not only in malignant lesions but also in benign lesions like papillary adenoma [16].

The diagnosis of collision tumor for these oncocytic and papillary lesions has clinical implications since the oncocytic part can be named chRCC and the papillary component can be papillary carcinoma type 1. The surgical approach, so far, to these mixed oncocytic lesions is partial nephrectomy or tumorectomy, and the patients are then followed to exclude recurrences. Our NGS analysis did not reveal any pathogenic mutations in any of the patients inour series. The majority of the detected mutations were variant of uncertain significance. In addition, given the allelic frequency being near 50%, they are strongly suspected ofgenetic polymorphisms. Based on the genetic profile and in accordance with our previous observations in clear-cell papillary kidney tumors, we can argue that these lesions are biologically benign [17]. A thorough observational study of these patients is needed to ascertain if they are also clinically benign and do not require follow-up.

## Figures and Tables

**Figure 1 diagnostics-11-00184-f001:**
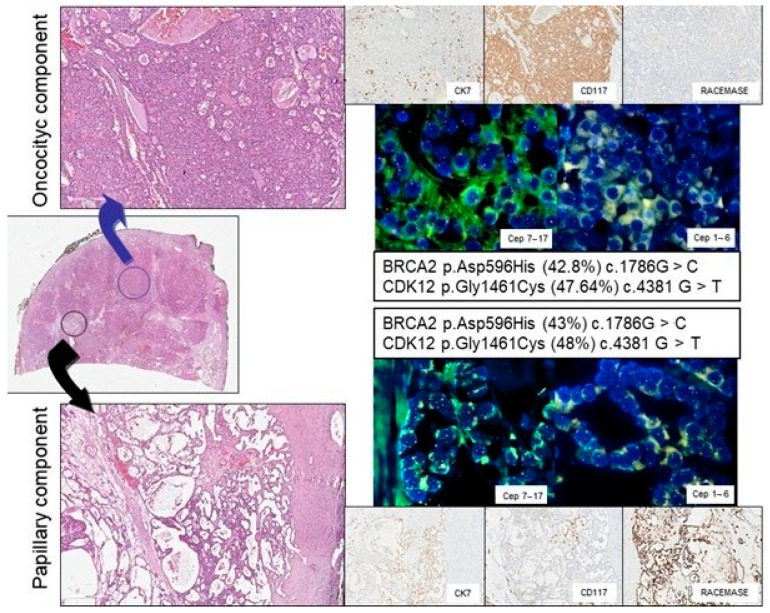
Representative histological, immunohistochemical, fluorescence in situ hybridization (FISH), and genetic analysis of a renal tumor with oncocytic (top panels) and papillary (bottom panels) features:magnification for hematoxylin and eosin, 10× and 20×; for immunohistochemistry, 20×; and for FISH, 100×.

**Table 1 diagnostics-11-00184-t001:** Clinical and pathological characteristics of the 11 cases.

Case	Sex	Age	Size	Side
1	M	66	3.5 cm	left kidney
2	F	42	3 cm	right kidney
3	F	49	2.5 cm	left kidney
4	F	80	NA	right kidney
5	M	45	3 cm	right kidney
6	M	68	2.6 cm	right kidney
7	M	49	2.6 cm	left kidney
8	M	56	8 cm	left kidney
9	F	65	3 cm	right kidney
10	F	56	6 cm	right kidney
11	M	72	3 cm	right kidney

**Table 2 diagnostics-11-00184-t002:** Histological, immunophenotypic, and genetic features of the 11 cases: IHC (immunohistochemistry), FISH (fluorescent in situ hybridization), and HOT (hybrid oncocytic tumor).

Case	Histology	IHC	FISH Profile	Mutated Genes at NGS
1	Oncocytoma	CK7(−) AMACR(−) CD117(+)	Disomy	*BRCA2 and CDK12*
	Papillary	CK7(+) AMACR(+) CD117(−)	Gain 7	*BRCA2 and CDK12*
2	HOT	CK7(−) AMACR(−) CD117(+)	Disomy	NA
	Papillary	CK7(+) AMACR(+) CD117(−)	Disomy	
3	Oncocytoma	CK7(−) AMACR(−) CD117(+)	Disomy	*NBN and MRE11*
	Papillary	CK7(+) AMACR(+) CD117(-)	Disomy	*NBN and MRE11*
4	HOT	CK7(−) AMACR(−) CD117(+)	Disomy	No geneticvariants
	Papillary	CK7(+) AMACR(+) CD117(−)	Disomy	No geneticvariants
5	Oncocytoma	CK7(−) AMACR(−) CD117(+)	Disomy	*CCND2, FANCI; FGFR4, RNF43*
	Papillary	CK7(+) AMACR(+) CD117(−)	Gain 7 and 17	*CCND2, FANCI; FGFR4, RNF43*
6	Oncocytoma	CK7(−) AMACR(−) CD117(+)	Disomy	NA
	Papillary	CK7(+) AMACR(+) CD117(−)	Disomy	
7	HOT	CK7(−) AMACR(−) CD117(+)	Loss 1 and 6	NA
	Papillary	CK7(+) AMACR(+) CD117(−)	Loss 1 and 6	
8	HOT	CK7(−) AMACR(−) CD117(+)	Disomy	*MET, SLX4*
	Papillary	CK7(+) AMACR(+) CD117(−)	Disomico	*MET, SLX4*
9	HOT	CK7(−) AMACR(−) CD117(+)	Disomy	No genetic variants
	Papillary	CK7(+) AMACR(+) CD117(−)	Gain 7 and 17	No genetic variants
10	HOT	CK7(−) AMACR(−) CD117(+)	Gain 1	*ERCC2, RNF43*
	Papillary	CK7(+) AMACR(+) CD117(−)	Gain 1	*ERCC2, RNF43*
11	Oncocytoma	CK7(−) AMACR(−) CD117(+)	Disomy	NA
	Papillary	CK7(+) AMACR(+) CD117(−)	Disomy	

## Data Availability

Data are available to public upon requests.

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
