# Peer review of "Renal Tumors with Oncocytic and Papillary Features: A Phenotypic and Genotypic Study"

_diagnostics, 2021, doi:10.3390/diagnostics11020184_

Round 1

Reviewer 1 Report

This work consists of a phenotypic and genotypic study of renal tumors with oncocytic and papillary features.

Overall the study design and methodology seem appropriate and the results and discussion are well presented. In addition to that, the study also adds advances in the field of renal cancer.

My only comment goes to the manuscript title. The authors wrote "Renal tumors with oncologic and papillary features.." and I think that they meant "oncocytic" instead of "oncologic", but please confirm and correct that. 

Author Response

We thank the reviewer for the appreciation of our work.

Sorry, the title was a mistyping. We have fixed this

Reviewer 2 Report

This paper by Franceschini et al. is a very interesting work aimed at the clarification of the features of those renal cell cancer with mixed morphology. The methods utilized are non only traditional morphology, but also molecular and modern next generation sequencing. They claim to change the terminology "collision tumors" for this kind of lesion and would encourage the use of a new terminology.

Overall the paper is well planned and well written, the results are sound and the discussion clear. 

In this Reviewer's opinion, the paper needs just minor revisions. 

In the introduction section (and then in the Reference section) it would be appreciated a mention as counterpart of the ref. 7, as a stimulus to plan and to complete the present study. In other words, The Authors should explain which previous study prompted them to check the Pathology lab archive to find those 11 cases object of their work to verify their phenotype and genotype.

Methods: the Authors state that they used IHC (CK7 and AMACR) to diagnose papillary component. What about cases eventually negative for CK7, as it occurs in type 2 tumors? Authors are encouraged to explain this issue. Finally, it is just a curiosity of this Reviewer to know why it has been utilized 12% as a cutoff: as it is stated that it has been an arbitrary choice, why 12 and not 13?

Results: have the Authors quantified the percentage of the 2 components? is there any correlation between cases with smaller benign component and the size of the tumors? It could depend, mostly for larger tumors, on eventual low amount of samples taken during the gross examination.

The discussion is sound and can be accepted as it is, mostly if queries about the results section will not change the result of the study. Otherwise, the Authors should complete the discussion in the light of the new data.

Author Response

We thank the reviewer for the appreciation of our work

Point-by-poit response to the comments:

In the introduction section (and then in the Reference section) it would be appreciated a mention as counterpart of the ref. 7, as a stimulus to plan and to complete the present study. In other words, The Authors should explain which previous study prompted them to check the Pathology lab archive to find those 11 cases object of their work to verify their phenotype and genotype.

The main reason why we started this study was to demonstrate that these tumors were indolent or benign despite the presence of two different counterparts at immunohistochemistry and FISH. In this revised version of the paper we added a sentence in the Introduction and another reference

Methods: the Authors state that they used IHC (CK7 and AMACR) to diagnose papillary component. What about cases eventually negative for CK7, as it occurs in type 2 tumors? Authors are encouraged to explain this issue. Finally, it is just a curiosity of this Reviewer to know why it has been utilized 12% as a cutoff: as it is stated that it has been an arbitrary choice, why 12 and not 13?

We decided to use CK7 and AMACR only since in all the cases of our series the papillary component was type I.

The cut off of 12% derives from a paper previously published by us (Giunchi et al. Pathology. 2016 Jan;48(1):41-6) where the mean Ki67 value of HOT tumors was 12% but the cases were just two and now we said here that the cut off was arbitrarily chosen.

Results: have the Authors quantified the percentage of the 2 components? is there any correlation between cases with smaller benign component and the size of the tumors? It could depend, mostly for larger tumors, on eventual low amount of samples taken during the gross examination.

We thank the reviewe for this correct observation. The amount of papillary component was scanty in all tumor compared to the oncocytic counterpart. In three cases the NGS was not accomplished (as stated in the Results) because the ppaillary counterpart was too scarce. As far as we can say in our cohort the size of the papillary component was not relevant.